# Study of the Comparative Effect of Sintering Methods and Sintering Additives on the Microstructure and Performance of Si_3_N_4_ Ceramic

**DOI:** 10.3390/ma12132142

**Published:** 2019-07-03

**Authors:** Liangliang Yang, Allah Ditta, Bo Feng, Yue Zhang, Zhipeng Xie

**Affiliations:** 1School of Materials Science and Engineering, Beihang University, Beijing 100191, China; 2State Key Laboratory of New Ceramics and Fine Processing, School of Materials Science and Engineering, Tsinghua University, Beijing 100084, China

**Keywords:** GPS, SPS, microstructure, mechanical performance, sintering additives, silicon nitride

## Abstract

The Si_3_N_4_ ceramics were prepared in this study by gas pressure sintering (GPS) and spark plasma sintering (SPS) techniques, using 5 wt.% Yb_2_O_3_–2 wt.% Al_2_O_3_ and 5 wt.% CeO_2_–2 wt.% Al_2_O_3_ as sintering additives. Based on the difference in sintering methods and sintering additive systems, the relative density, phase composition, phase transition rate, microstructure, mechanical properties, and thermal conductivity were comparatively investigated and analyzed. SPS proved to be more efficient than GPS, producing higher relative density, bending strength, hardness, and thermal conductivity of Si_3_N_4_ ceramic with both additive systems; however, the phase transition rate and fracture toughness were lower. Similarly, higher bending strength, hardness, and thermal conductivity were achieved with Yb_2_O_3_–Al_2_O_3_ than CeO_2_–Al_2_O_3_ in the case of GPS and SPS, and only the relative density, fracture toughness, and phase transition rate were lower.

## 1. Introduction

Silicon nitride is an attractive structural material with high-temperature strength, good hardness, and excellent wear and corrosion resistance, and it has wide applications in the high-temperature structural material field, such as for cutting tools, bearings, high-pressure plugs, and sealing rings.

The research on silicon nitride mainly consists of the following: rational selection of sintering aids, sintering techniques and control of technological parameters, manipulation of microstructure (such as grain boundary phase, grain boundary thickness, grain size, crystal phase transition rate from α-Si_3_N_4_ to β-Si_3_N_4_, etc.) to accordingly adjust the mechanical and thermal properties. Silicon nitride has a strong covalent bond structure. Therefore, the diffusion coefficient is low, and the sintering driving force is weak. It is challenging to densify silicon nitride without sintering aids; thus, choosing a suitable additive for densification is a crucial step. The selection criteria while sintering for sintering aids, in general, are as follows: (1) capable of forming liquid phase to promote densification and crystal phase transition from α-Si_3_N_4_ to β-Si_3_N_4_ by solution reprecipitation; (2) full or partial crystallization of the liquid phase upon cooling to reduce the proportion of glassy phase at grain boundaries; (3) enough high-temperature strength of grain boundary phase [1]. At present, the usual additives are Y_2_O_3_–Al_2_O_3_, Y_2_O_3_–Nd_2_O_3_, and MgO–SiO_2_ [2,3,4,5] for liquid phase formation and good densification. However, the crystallization of the second phase at grain boundaries is incomplete. In recent years, researchers began to notice Yb_2_O_3_ as a preferable sintering additive because Yb_2_O_3_ not only has a similar sintering effect to Y_2_O_3_, but the bending strength of the silicon nitride with Yb_2_O_3_ is also better than that of Y_2_O_3_ [6,7].

Kondo et al. [8,9] added 13 wt.% Yb_2_O_3_ into silicon nitride and obtained superior mechanical properties at room and high temperature (1773 K). Vetrano et al. [10,11] systematically investigated the influence of Yb_2_O_3_ content on mechanical properties for silicon nitride and found that, when the content of Yb_2_O_3_ was higher than 5 wt.%, it significantly improved the crystallization of grain boundary phases and, consequently, material strength. Research on CeO_2_ also established it as an effective additive which promotes liquid phase sintering at lower temperature, while remarkably increasing the bending strength and hardness of silicon nitride [12,13]. Other documents reported that Al_2_O_3_ can promote the mechanical properties of silicon nitride ceramics [14,15]. Thus, we infer that it can densify Si_3_N_4_ ceramics with the addition of Yb_2_O_3_–Al_2_O_3_ or CeO_2_–Al_2_O_3_, and such ceramics should have high thermal conductivity and excellent high-temperature mechanical properties.

The industries mainly adopt gas pressure sintering (GPS) for silicon nitride as it is an economical method; however, the technique normally consists of a prolonged heating cycle. On the other hand, many research institutes apply the spark plasma sintering (SPS) technique because it is a relatively new sintering technique that can rapidly densify ceramic powders at lower temperatures; however, this equipment is expensive. Thus, how to strike a good balance between the ceramic performance and the cost is a critical issue. Furthermore, the comparative effects of sintering additives Yb_2_O_3_–Al_2_O_3_ and CeO_2_–Al_2_O_3_ using GPS and SPS sintering techniques are less studied for silicon nitride. In this work, we used 5 wt.% Yb_2_O_3_–2 wt.% Al_2_O_3_ and 5 wt.% CeO_2_–2 wt.% Al_2_O_3_ as sintering additives and employed GPS and SPS to prepare silicon nitride ceramics. The aim was to comparatively study the influence of sintering techniques and sintering additives on the structure and performance of silicon nitride. The effects on phase transformation, grain boundary phase, bending strength, hardness, fracture toughness, microstructure, and thermal conductivity of the ceramics were investigated.

## 2. Experimental Procedure

### 2.1. Raw Material and Preparation of Samples

Commercial Si_3_N_4_ powder (average size of ~0.1μm, purity ≥ 99%, α phase content ≥ 95 wt.%, dissociated Si ≤ 0.3 wt.%, Fe ≤ 0.25 wt.%, Jinsheng Ceramics Co., Ltd., Changzhou, China) and additives Yb_2_O_3_, CeO_2_, and Al_2_O_3_ (analytical grade) (Guoyao, Beijing, China) were used as the starting materials. The samples with an additive combination of Yb_2_O_3_–Al_2_O_3_ and CeO_2_–Al_2_O_3_ were named YA and CA, respectively, while, to identify the sintering technique, prefixes S and G were defined before YA and CA to represent SPS and GPS, respectively. The powders were mixed in ethanol (Guoyao, Beijing, China) in accordance with Table 1, ball-milled for 20 h (h), vacuum-dried, and subsequently sieved (75 μm).

For SPS, the powder mixture was loaded into a graphite mold and sintered at 1873 K for 6 min with a maximum applied pressure of 50 MPa. Similarly, for GPS, the samples were firstly dry-press molded, before being cold isostatic pressed at 200 MPa for 3 min, and finally sintered at 2053 K for 60 min with maximum applied nitrogen pressure of 6 MPa. After SPS and GPS sintering, the ceramic blocks were cut, ground, and polished for physical analysis and characterization.

### 2.2. Characterization

The bulk density of the samples was measured in distilled water by the Archimedean displacement technique. The relative density was calculated by dividing the bulk density with theoretical density. The bending strength of the samples was determined by three-point bending. The samples with the size of 1.5 mm × 1.5 mm × 20 mm were prepared by grinding and polishing, and the tests were carried out on an Instron-5500 machine with a span width of 16 mm and a loading rate of 0.5 mm/min. A total of 10 samples were employed for the testing, and the average value was determined. The hardness was determined using a microhardness tester (Model: TUKON^TM^ 2500, Wilson, New York, USA) on the polished surface of the samples with a load of 5 kg and dwell time of 15 s. The indentation fracture resistance was measured with a load of 20 kg using the same equipment and calculated according to the Shetty equation [16]. A total of 10 indentations were made on the surface of each sample for microhardness and fracture toughness testing. The phase composition was determined by X-ray diffraction (XRD; D8 ADVANCE A25 BRUKER, BRUKER, Karlsruhe, Germany) using Cu-Kα radiation. A scanning electron microscope (SEM; S-4800, hitachi, Tokyo, Japan) was employed to observe the microstructure of fractured samples. The thermal conductivity of the samples was calculated by Equation (1) [17].
(1)$$\lambda =\rho \cdot C_{p}\cdot \alpha ,$$
where *ρ*, α, and *C_P_* are the bulk density, thermal diffusivity, and heat capacity, respectively. Thermal diffusivity (α) was determined by the wave thermal analysis method (Model: ai-phase mobile 1, Japan) on relevant samples with a thickness of about 0.5 mm, while a constant value of heat capacity, 680 J/(kg·K) was used for the calculations of thermal conductivity.

## 3. Results and Discussion

The specific sintering technique and particular additive combinations had characteristic effects on the performance of Si_3_N_4_ ceramics, as discussed below.

### 3.1. Relative Density

The effect of sintering additive combinations and sintering techniques on the relative density of the sintered samples is shown in Table 2. Irrespective of the sintering additives, the density of SPS sintered samples was greater than that of GPS sintered samples, mainly because the higher pressure of 50 MPa was applied in SPS sintering, which promoted densification [18]. Under the same sintering technique, the relative density of CA was slightly higher than that of YA because CeO_2_ can form a eutectic liquid phase (CeO_2_–SiO_2_/CeO_2_–SiO_2_–Si_3_N_4_) at lower temperatures than Yb_2_O_3_ [19,20]. With the rise of temperature to respective sintering temperatures, the viscosity of the eutectic liquid phase further reduced, which effectively enhanced the rearrangement of silicon nitride granules and promoted densification.

### 3.2. Phase Composition

The XRD profile patterns of the sintered samples are presented in Figure 1. It can be observed that different sintering techniques and additive combinations had dissimilar effects on the phase composition of the samples. The main crystalline phase in all the samples was β-Si_3_N_4_. The XRD patterns of SPS sintered samples in comparison to GPS sintered samples showed some peaks of α-Si_3_N_4_, indicating that the transformation of α-Si_3_N_4_ to β-Si_3_N_4_ in SPS sintering was not entirely completed. SPS sintering is a rapid sintering process and, therefore, time may not be sufficient for the complete phase transformation of α-Si_3_N_4_ to β-Si_3_N_4_. However, in the case of GPS sintered samples, the phase transformation from α-Si_3_N_4_ to β-Si_3_N_4_ was almost entirely completed because of the higher sintering temperature and prolonged heat preservation time.

In the samples sintered with the Yb_2_O_3_–Al_2_O_3_ additive combination, some peaks of crystalline grain boundary phase Yb_8_Si_4_N_4_O_14_ were detected, which could have formed upon cooling of the eutectic liquid produced by the reaction of Yb_2_O_3_ with SiO_2_ on the surface of Si_3_N_4_ and Si_3_N_4_ itself. The intensity of this phase increased with the rise of sintering temperature to 2073 K in GPS. The characteristic feature of Yb_2_O_3_ forming a crystalline grain boundary phase was already reported by researchers [10].

### 3.3. Phase Transition Rate

The schematic sequence of Si_3_N_4_ crystal phase transition is shown in Figure 2. With the formation of the liquid phase by the reaction of sintering additives with SiO_2_ on the surface of Si_3_N_4_ and partly Si_3_N_4_ itself, the α-Si_3_N_4_, being an unstable phase, dissolved in the liquid phase and re-precipitated in the form of elongated β-Si_3_N_4_ needles. The initial β-phase seeds provide nucleation sites or promote the re-precipitation process. The aspect ratio of newly produced or re-precipitated β-phase depends on the sintering parameters, sintering additives, the viscosity of liquid phase, etc.

The phase content of β-Si_3_N_4_ was determined by the calculation method proposed by Gazzara and Messier [21], as shown in Equation (2).
(2)$${\beta –\mathrm{Si}}_{3}N_{4}\left(\mathrm{wt}\%\right)=\frac{I_{\beta \left(101\right)}{+I}_{\beta \left(210\right)}}{\left[I_{\alpha \left(102\right)}{+I}_{\alpha \left(210\right)}{+I}_{\beta \left(101\right)}{+I}_{\beta \left(210\right)}\right]},$$
where I_β(101)_ and I_β(210)_ are the diffraction intensities of the (101) and (210) planes of β-Si_3_N_4_, and I_α(102)_ and I_α(201)_ represent the intensities of the (102) and (201) planes of α-Si_3_N_4_.

For both additive combinations, the phase transition rate, α-Si_3_N_4_ to β-Si_3_N_4_, for GPS was higher than for SPS, as shown in Table 2. The sintering temperature of GPS was higher, and the holding time at the sintering temperature was also more than one hour compared to the lower sintering temperature and shorter holding time (15 min) in SPS sintering. Therefore, the higher sintering temperature and prolonged holding time might have facilitated the higher phase transition in GPS.

The variation in phase transition rates of YA and CA additive combinations in GPS sintering was not noticeable, and both were close to 99.6%. However, SPS sintering had a small difference in which the phase transition rate of CA was slightly higher than that of YA. The reason may be the lower eutectic liquid formation temperature (1723 K) [22] with CeO_2_ via the CeO_2_–SiO_2_–Si_3_N_4_ reaction, which offers more time for α-Si_3_N_4_ dissolution in the eutectic liquid phase.

### 3.4. Microstructure

SEM micrographs of the fractured surfaces of SYA, SCA, GYA, and GCA samples are shown in Figure 3. The microstructures were mainly dominated by the long cylindrical β-phase. Pores were almost entirely excluded, ensuring densification of the Si_3_N_4_ samples. From Figure 3a,b, it can be seen that the microstructure of samples sintered with SPS sintering was fine. The thickness of the elongated β-grains was small, and the grains were tightly interconnected in comparison to GPS sintered samples (Figure 3c,d), which comprised a coarse microstructure. Similarly, the samples sintered with the YA additive combination (Figure 3a,c) contained comparatively less of an intergranular phase than the samples with the CA additive combination (Figure 3b,d).

### 3.5. Mechanical Properties

#### 3.5.1. Bending Strength

The bending strength of the Si_3_N_4_ samples is shown in Figure 4. The bending strength values of SPS sintered samples were higher than those of GPS sintered samples. The bending strength values of the SYA and SCA samples were 1013 MPa and 954 MPa, respectively; however, with GPS sintering, the corresponding values were only 915 MPa (GYA) and 887 MPa (GCA). The fine microstructure with uniform distribution of elongated β-grains, as shown in Figure 3a,b, resulted in higher bending strength values in SPS sintered samples.

The bending strength of samples sintered with the YA additive was greater than that with CA in both GPS and SPS sintering. According to the calculation method proposed by Tseng shown in Equation (3) [23], the bending strength depends on the average length-to-diameter ratio, called the aspect ratio (AR), of β-Si_3_N_4_ grains. It can be observed from Figure 3a,c that the aspect ratio of β-Si_3_N_4_ grains with the YA additive combination was higher than that with CA, indicating that YA is more useful to promote the bending strength than CA.
(3)$$\mathrm{AR}=\frac{L}{W}=\left(\frac{K_{L}^{1/3}}{K_{W}^{1/5}}\right){\times t}^{2/15}\exp\left[\frac{\left(\frac{Q_{L}}{3}-\frac{Q_{W}}{5}\right)}{\mathrm{kT}}\right],$$
where AR is the aspect ratio, K_L_ and K_W_ are the growth rate constants of β-Si_3_N_4_ in length and width, respectively, Q_L_ and Q_W_ are the respective activation energies (Q_L_ = 686 J/mol, Q_W_ = 772 J/mol), k is the gas constant, t is the time, and T is the absolute temperature.

#### 3.5.2. Hardness

The hardness values of the sintered silicon nitride samples are shown in Figure 5. The hardness of ceramic materials depends on the bonding type, crystal structure, and chemical composition. In addition, the microstructure, cracks, and impurities of ceramic materials have an influence on the hardness.

The hardness of the samples sintered with the YA additive combination was higher than that with CA. Xie [24] illustrated that smaller-radii rare-earth ions offer higher scratch and penetration resistance and, accordingly, demonstrate higher hardness. The atomic radius of Ce is greater than that of Yb; therefore, conversely, the hardness of the samples with the YA additive combination is greater than that of CA. Research proved [25] that the hardness of sintered Si_3_N_4_ depends on the content of α-Si_3_N_4_, density, and characteristics of the intergranular glassy phase. Therefore, a higher content of α-Si_3_N_4_, high density, and large glass hardness are beneficial for improving the hardness of Si_3_N_4_. From Table 2, it can be seen that the content of α-Si_3_N_4_ phase in the samples which were sintered by SPS was more substantial than those sintered by GPS. The samples sintered by GPS (Table 2) were composed almost entirely of the β-Si_3_N_4_ phase without α-Si_3_N_4_ and a small amount of crystalline phase Yb_8_Si_4_N_4_O_14_. Moreover, the relative density of the silicon nitride samples with SPS was greater than that of the samples with GPS. Therefore, the hardness after SPS sintering was greater than GPS sintering, which is consistent with the plot shown in Figure 5.

#### 3.5.3. Fracture Toughness

The fracture toughness of GPS and SPS sintered samples with YA and CA additive combinations is shown in Figure 6. 

It can be seen that the fracture toughness values of the SPS sintered samples were greater than those of GPS sintered samples. With the YA and CA additive combinations, the values by SPS sintering were 9.5 MPa·m^1/2^ and 8.6 MPa·m^1/2^, respectively; however, the corresponding values by GPS sintering were 8.9 MPa·m^1/2^ and 8.5 MPa·m^1/2^, respectively. Fracture toughness was observed to enhance when large elongated grains are formed in a fine matrix [26]. Therefore, the fine microstructure in SPS sintered samples (Figure 3a,b) improved the fracture toughness. 

Similarly, the fracture toughness of samples with the YA additive was greater than that with CA in SPS and GPS sintering, indicating that the YA system can promote fracture toughness more than the CA system. 

Fracture mode has an influence on the fracture toughness. The long columnar β-Si_3_N_4_ grains in the sintered Si_3_N_4_ ceramics influence the crack propagation path. It means that the crack propagation takes on more tortuous paths, which consume extra fracture energy. Therefore, in order to find the crack propagation behavior and toughening mechanism of Si_3_N_4_ ceramic at room temperature, the crack propagation paths of GPS sintered samples with the YA and CA additive systems were observed under the scanning electron microscope. The microscopic appearance of the fractured samples is shown in Figure 7; it can be assumed that crack propagation in the CA additive system mainly adopted a transgranular mode, while that with YA mainly focused on the intergranular mode of fracture.

The specific interlocking microstructure can effectively improve the fracture toughness of Si_3_N_4_ ceramic via the elongated β-Si_3_N_4_ grains, due to crack deflection and crack bridging.

The Si_3_N_4_ ceramics belong to the class of hard brittle materials. The toughening method is mainly through crack bridging and deflection by elongated β grains and the toughening behavior of the grain pulling effect. On the one hand, the crack path along the grains induces the propagation of the crack with more twists and turns. On the other hand, Si_3_N_4_ ceramics produce more bifurcation, crack deflection, and a toughening effect. For instance, grain pulling is the main cause of the increase in fracture toughness.

Murgatroyd et al. [27] believes that there are some elastic regions in the glass that contain small areas of quasi-viscous material. According to Marsh et al. [28], glass is not a pure brittle material, but is treated as an elastic–plastic complex. However, more experiments are required to verify these claims.

### 3.6. Thermal Conductivity

The thermal conductivity with the YA additive was higher than that with CA as shown in Table 2. The possible explanation can be given by the formula derived by Morikawa [24,29].
(4)$$\mathrm{lg}\lambda =(\mathrm{lg}\lambda_{m}-C_{f}\mathrm{lg}\lambda_{f})V+C_{f}\mathrm{lg}\lambda_{f},$$
where λ is the net thermal conductivity of Si_3_N_4_ ceramic, while *λ_m_* and *λ_f_* are representations of the thermal conductivities of intergranular or intercrystalline phase and β-Si_3_N_4_ columnar crystals, respectively. C*_f_* is the thermal resistance effect and *V* is the intercrystalline volume fraction. Therefore, the thermal conductivity of silicon nitride ceramics, i.e., lg*λ*, is proportional to the intercrystalline phase volume fraction *V*. 

Related reports [30] about the sintering processing of AlN ceramic elaborated that, in the presence of a carbon and nitrogen atmosphere, Y–Al– meets composite oxide from inside the base, and runs to the substrate surface to generate YN and AlN, resulting in AlN lattice purification, thereby improving the thermal conductivity of AlN substrate. A similar situation may explain that the experiment with intercrystalline Yb_8_Si_4_N_4_O_14_ may partly be broken down into Yb_2_O_3_ and Si_3_N_4_, and, under the effect of the concentration gradient diffusion, it may transfer gradually to the sample surface, thereby reducing the content of intergranular phase. The thermal conductivity of YA was greater than CA in the SPS or GPS sintering method.

According to Equation (5) by Slack [31], the average atomic weight of the β-Si_3_N_4_ strong atomic bond and non-harmonic vibration of the crystal are similar to SiC and AlN. Therefore, the content of β-Si_3_N_4_ is proportional to its thermal conductivity, and it has some influence on grain boundary, grain, and so on. Therefore, the thermal conductivity of GPS sintering samples is higher than that of SPS.
(5)$$k=\frac{B\bar{M}\delta^{3}\Theta_{D}}{T\gamma^{2}},$$
where $\bar{M}$ is the mean atomic weight, δ is the cube root of the atomic volume of a unit, γ is the Gruneisen constant, $\Theta_{D}$ is the Debye temperature, and B is a constant.

High-thermal-conductivity ceramics should have the following conditions: (1) small average atomic weight; (2) high atomic bonding strength; (3) relatively simple crystal structure; and (4) low lattice anharmonic vibration. However, the thermal conductivity of silicon nitride was not high in this experiment, due to its complicated structure, which has a significant relationship with the scattering of phonons.

## 4. Conclusions

The dense and uniform Si_3_N_4_ ceramics were prepared by GPS and SPS using 5 wt.% Yb_2_O_3_–2 wt.% Al_2_O_3_ and 5 wt.% CeO_2_–2 wt.% Al_2_O_3_ additive systems. The relative density, phase composition, phase transition rate, mechanical properties, and thermal conductivity were comparatively studied based on the sintering techniques and additive systems. The results are summarized below.

For GPS, the bending strength, hardness, and thermal conductivity of the YA system were higher than that of the CA system, while the fracture toughness and phase transformation rate were lower. Furthermore, the highest value of fracture toughness (9.5 MPa·m^1/2^) was also obtained in the case of the CA additive system. For SPS, the highest values of bending strength (1013 MPa), Vickers hardness (18.4 GPa), and thermal conductivity (49 W/(m·K)) were obtained in the case of the YA additive system; conversely, the relative density, phase transition rate, and the fracture toughness of the CA additive system were greater than that with YA. However, irrespective of the employed additive systems, the relative density, thermal conductivity, bending strength, and Vickers hardness of the SPS sintered samples were higher than those of the GPS sintered samples.

Further research studies will mainly focus on investigating the sintering mechanism, with impurity phase reduction, and a detailed chemical analysis of sintered Si_3_N_4_ ceramics. 

## Figures and Tables

**Figure 1 materials-12-02142-f001:**
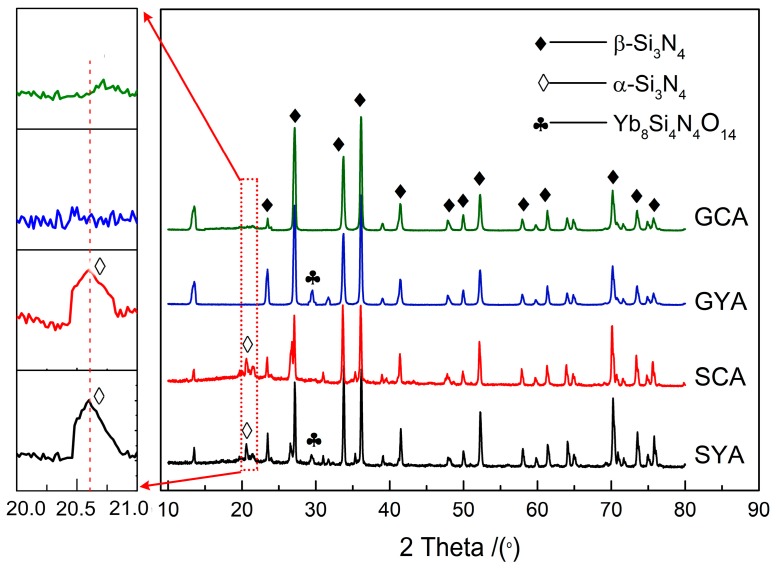
X-ray diffraction (XRD) patterns of the samples obtained by SYA, SCA, GYA, and GCA, where S represents spark plasma sintering (SPS), G represents gas pressure sintering (GPS), CA represents CeO_2_–Al_2_O_3_, and YA represents Yb_2_O_3_–Al_2_O_3_.

**Figure 2 materials-12-02142-f002:**
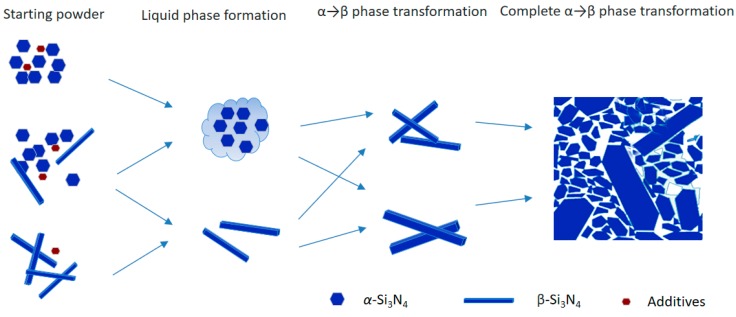
The schematic representation of Si_3_N_4_ crystal phase transition.

**Figure 3 materials-12-02142-f003:**
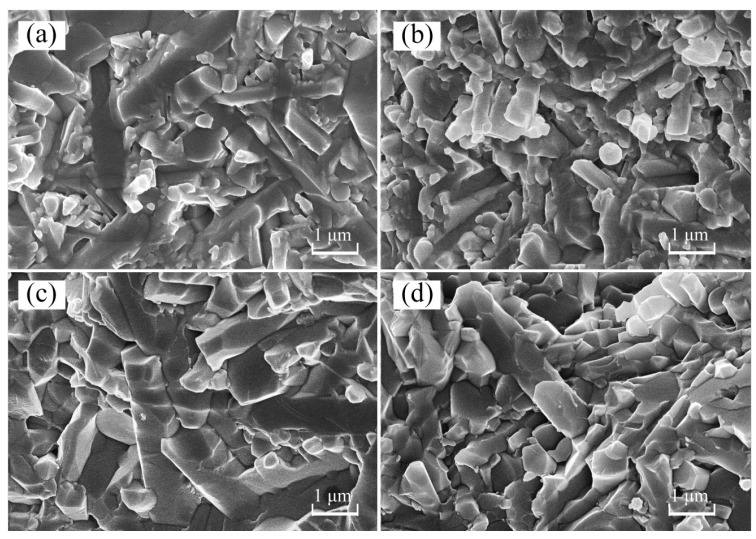
SEM micrographs of the fractured surfaces of Si_3_N_4_ samples: (**a**) SYA; (**b**) SCA; (**c**) GYA; (**d**) GCA.

**Figure 4 materials-12-02142-f004:**
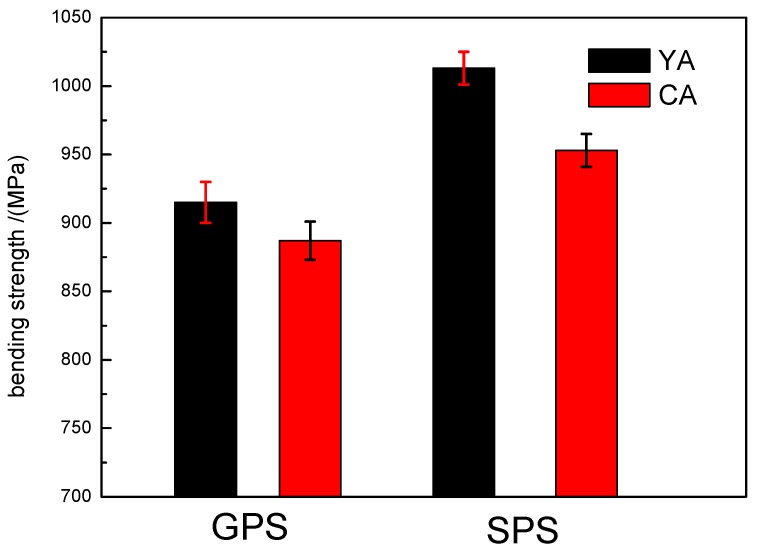
Bending strength (MPa) of GPS and SPS sintered silicon nitride samples.

**Figure 5 materials-12-02142-f005:**
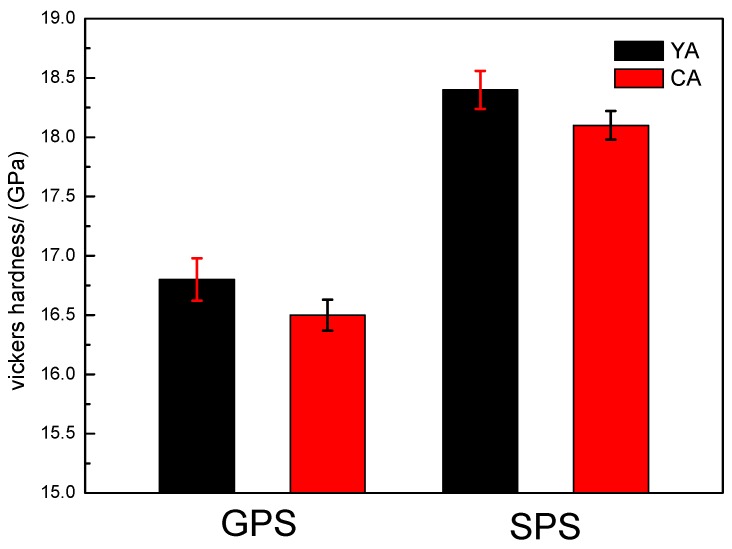
Micro Vickers hardness (GPa) of GPS and SPS sintered silicon nitride samples.

**Figure 6 materials-12-02142-f006:**
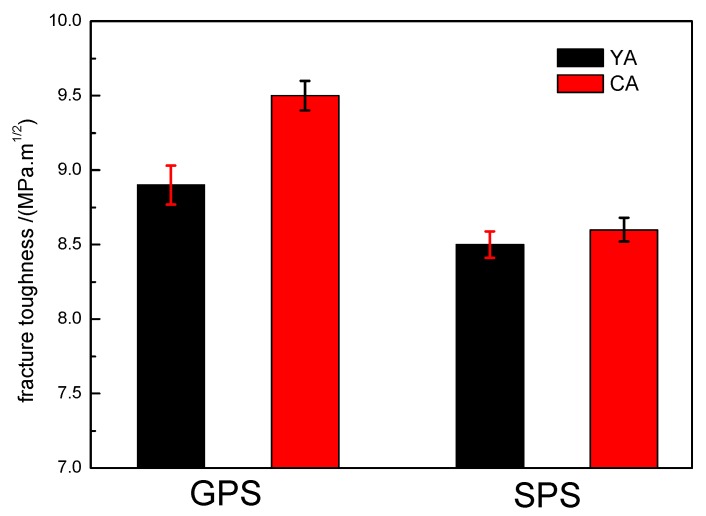
Fracture toughness (MPa·m^1/2^) of GPS and SPS sintered silicon nitride samples.

**Figure 7 materials-12-02142-f007:**
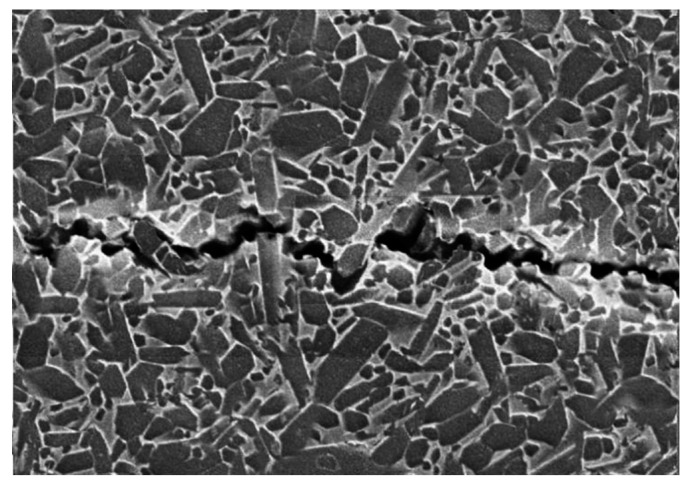
GCA crack propagation micromorphology.

**Table 1 materials-12-02142-t001:** Experimental parameters. S—spark plasma sintering (SPS); G—gas pressure sintering (GPS); CA—CeO_2_–Al_2_O_3_; YA—Yb_2_O_3_–Al_2_O_3_.

Samples	Sintering Conditions Sintering Technique/Sintering Temperature/Holding Time	Yb_2_O_3_ (wt.%)	CeO_2_ (wt.%)	Al_2_O_3_ (wt.%)	Si_3_N_4_ (wt.%)
SYA	SPS/1873 K/6 min	5	0	2	93
SCA		0	5	2	93
GYA	GPS/2053 K/60 min	5	0	2	93
GCA		0	5	2	93

**Table 2 materials-12-02142-t002:** The observed density, β-phase percentage, and thermal conductivity of samples.

Sample	β/(α + β) (wt.%)	Bulk Density (kg/m^3^)	Relative Density (%)	Thermal Conductivity (W/(m·K))
SYA	91.43	3310	99.6	49
SCA	93.25	3370	99.7	38
GYA	99.58	3290	99.2	42
GCA	99.61	3350	99.3	35

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
