# Peer review of "Study of the Comparative Effect of Sintering Methods and Sintering Additives on the Microstructure and Performance of Si3N4 Ceramic"

_materials, 2019, doi:10.3390/ma12132142_

Round 1
Reviewer 1 Report
In this study, the authors compare Si3N4 ceramics prepared by two different methods (SPS and GPS) using Yb2O3, CeO2 and Al2O3 as sintering additives. However, the results of XRD analysis must be thoroughly reinterpreted before publication:
1) The intensities of the diffraction peaks in the XRD patterns SCA and SYA (Fig.1) are much less than that of the patterns GCA and GYA. The intensity axis of the patterns SCA and SYA should be stretched to its normal size!
2) The XRD patterns clearly show the peaks of impurity phases: a series of the peaks positioned at 2Q = 19-22 (GCA, SYA and SCA), a sharp peak near 2Q =29 (SCA and SYA). It therefore cannot be claimed that a pure phase was produced by SCA, SYA and GCA methods.
3) The authors have identified a-Si3N4 phase in SCA and SYA samples by referring to a single peak near 2Q = 27, which is insufficient evidence, especially considering that the XRD patterns show other impurity peaks (see para. 2 above). The authors should provide the crystallographic database reference number (ID or PDF number) for a-Si3N4 that they used for phase identification.
4) Similarly, the authors suggested the presence of Yb8Si4N4O14 phase by referring to a single peak near 2Q = 72. However, the entire series of the peaks in this 2Q region can be attributed to b-Si3N4 phase (see figure in reviewer`s attachments). According to the standard XRD pattern PDF 51-0340, the major peaks of Yb8Si4N4O14 are positioned at 2Q = 29,36 and 31.71, but this is not the case here.
In view of the above, the whole discussion in section «3.3. Phase Transition Rate» is doubtful.

Reviewer 2 Report
Dear Sir,
The paper is interesting and well prepared however it can be improved in some area as follows:
In the abstract
· It is not recommended to use abbreviation at the beginning thus please indicate them to the reader: GPS and SPS.
- Some grammatical error should be corrected and there is some typos in this section for example : line 15: please change it: Prove to be more efficient.
- keywords should be arranged alphabetically.
Introduction section:
- Line 26: please add the word and before sealing ring.
- Line 32 please use thus instead of so
- Line 44: space is needed after reference [9,10]
- Line 55: should be SPS not GPS
- LINE 58: please add issue to critical at the end of the sentence.
- Line 59: difficult to be understood by the reader.
In the other sections
- Caps and the format of the tables and figures: Please follow the author’s guide.
- Please use the SI unit.
- State the model of the any instrument used in the research.
- It is recommended to preform EDX and XPS analysis for the samples to obtain more information about the chemical species on the surface of the obtained ceramics.
- Fig.3 is not in a good quality and cannot be distinguished if it printed in black and white.
- Conclusion need to be modified and future work should be added at the end of it.
· Reference should be checked carefully according to the style which is specified by the journal in particular reference no. 7 and 14.
Regards
Round 2
Reviewer 1 Report
Accept in present form
Reviewer 2 Report
Dear Sir,
thanks for the modification I wish you move forward for the further investigation in this interesting topic.
Regards